# Low-Dose Delta-9-Tetrahydrocannabinol as Beneficial Treatment for Aged APP/PS1 Mice

**DOI:** 10.3390/ijms23052757

**Published:** 2022-03-02

**Authors:** Yanhong Wang, Yuzhu Hong, Jiyu Yan, Breanna Brown, Xiaoyang Lin, Xiaolin Zhang, Ning Shen, Minghua Li, Jianfeng Cai, Marcia Gordon, David Morgan, Qingyu Zhou, Chuanhai Cao

**Affiliations:** 1Department of Pharmaceutical Sciences, Taneja College of Pharmacy, University of South Florida, Tampa, FL 33612, USA; yanhongwang@usf.edu (Y.W.); yuzhu1@usf.edu (Y.H.); jiyuyan@usf.edu (J.Y.); breannabrown@knights.ucf.edu (B.B.); xlin3@usf.edu (X.L.); ningshen@usf.edu (N.S.); minghua@usf.edu (M.L.); 2Byrd Alzheimer’s Center and Research Institute, Morsani College of Medicine, University of South Florida, Tampa, FL 33613, USA; xiaolin1@usf.edu; 3Department of Chemistry, College of Arts & Sciences, University of South Florida, Tampa, FL 33620, USA; jianfengcai@usf.edu; 4Department of Translational Neuroscience, College of Medicine, Michigan State University, Grand Rapids, MI 49503, USA; mngordon@msu.edu (M.G.); scientist.dave@gmail.com (D.M.)

**Keywords:** delta-9-tetrahydrocannabinol, cannabidiol, Alzheimer’s disease, radial arm water maze test, amyloid-β, APP/PS1 transgenic mice

## Abstract

Studies on the effective and safe therapeutic dosage of delta-9-tetrahydrocannabinol (THC) for the treatment of Alzheimer’s disease (AD) have been sparse due to the concern about THC’s psychotropic activity. The present study focused on demonstrating the beneficial effect of low-dose THC treatment in preclinical AD models. The effect of THC on amyloid-β (Aβ) production was examined in N2a/AβPPswe cells. An in vivo study was conducted in aged APP/PS1 transgenic mice that received an intraperitoneal injection of THC at 0.02 and 0.2 mg/kg every other day for three months. The in vitro study showed that THC inhibited Aβ aggregation within a safe dose range. Results of the radial arm water maze (RAWM) test demonstrated that treatment with 0.02 and 0.2 mg/kg of THC for three months significantly improved the spatial learning performance of aged APP/PS1 mice in a dose-dependent manner. Results of protein analyses revealed that low-dose THC treatment significantly decreased the expression of Aβ oligomers, phospho-tau and total tau, and increased the expression of Aβ monomers and phospho-GSK-3β (Ser9) in the THC-treated brain tissues. In conclusion, treatment with THC at 0.2 and 0.02 mg/kg improved the spatial learning of aged APP/PS1 mice, suggesting low-dose THC is a safe and effective treatment for AD.

## 1. Introduction

Delta-9-tetrahydrocannabinol (THC) and cannabidiol (CBD) are the two main phytocannabinoids found in plants of the Cannabis genus [1]. Although THC has exactly the same chemical formula as CBD (i.e., C_21_H_30_O_2_), there is a slight difference in their atomic arrangement in that THC contains a cyclic ring, whereas CBD contains a hydroxyl group (Appendix A) [1,2]. Due to the difference in chemical structure, THC and CBD are found to have different pharmacological effects. THC is considered the main psychotropic constituent of cannabis, acting as a partial agonist at cannabinoid type 1 (CB1) and type 1 (CB2) receptors of the endocannabinoid system [3]. CBD is non-psychotropic and interacts with many receptors and proteins other than the CB1 and CB2 receptors in the body [2]. Currently, the synthetically produced THC, dronabinol (Marinol^®^), has received FDA approval to treat nausea and vomiting caused by cancer chemotherapy. Plant-based pharmaceutical grade CBD (Epidiolex^®^) has been approved by the U.S. FDA for the treatment of seizures associated with Lennox–Gastaut syndrome and Dravet syndrome. Given the specific effect of THC on the endocannabinoid system and the diverse receptor profile of CBD, both THC and CBD have been implicated as potential neuroprotectants for mental and motor dysfunctions in neurodegenerative diseases including Alzheimer’s disease (AD) and Parkinson’s disease (PD) [4,5].

Alzheimer’s disease (AD) is the most common neurodegenerative disorder affecting over five million Americans and the annual direct medical costs have exceeded $300 billion (https://www.alz.org/aaic/downloads2020/2020_Facts_and_Figures_Fact_Sheet.pdf accessed on 1 March 2022). The incidence of AD increases exponentially with age and is shown to be similar for males and females [6]. Even though AD is the sixth leading cause of death in the United States, there is no treatment available to halt or cure AD (https://www.nia.nih.gov/health/alzheimers-disease-fact-sheet accessed on 1 March 2022). Neurologically, AD is characterized by the extracellular deposition of β-pleated assemblies of the amyloid β (Aβ) peptide in the form of diffuse plaques and neuritic plaques as well as the intracellular aggregation of insoluble hyperphosphorylated tau protein into neurofibrillary tangles (NFTs) within the perikarya of neurons [7,8]. These changes cause progressive memory loss as well as cognitive and behavioral impairments that ultimately lead to dementia. Mounting evidence demonstrates that Aβ accumulation can start as early as age 40 [9] and the major toxic isoform linked to neuron death is oligomeric Aβ [10,11]. Therefore, it is conceivable that the prevention or reduction in oligomeric Aβ formation may delay the onset or progression of AD.

Of the two main phytocannabinoids in cannabis, CBD has been extensively assessed for its potential therapeutic effect on AD in a number of preclinical studies [12]. In in vitro studies, CBD was found to protect against Aβ neurotoxicity [13], reduce Aβ production by promoting amyloid precursor protein ubiquitination [14], inhibit Aβ-induced tau protein hyperphosphorylation [15], modulate microglial cell functions [16], and improve cell viability [17]. In transgenic mouse models of AD, treatment with CBD alone was shown to prevent the development of social recognition memory deficits without influencing the anxiety parameter [18]. Results of in vivo studies using the Aβ-inoculated naïve mouse and rat models of AD-related neuroinflammation suggest that the neuroprotective effect of CBD is partly attributable to its anti-inflammatory property [19,20]. Treatment with CBD in combination with THC has been shown to improve memory deficits and prevent learning deficits in APP/PS1 transgenic mice that express a chimeric mouse/human amyloid precursor protein (Mo/HuAPP695swe) and a mutant human presenilin 1 (PS1-dE9) [21,22] and to improve certain neurological deficits in parkin-null and human tau overexpressing PK^−^/^−^/TauVLW mice [23]. In APP/PS1 mice, combined CBD and THC treatment reduced the glial reactivity associated with aberrant Aβ deposition through normalizing the pre-synaptic synaptosome associated protein 25 (SNAP25), decreasing the expression of glutamate receptors 2 and 3 (GluR2/3), and increasing the expression of γ-aminobutyric acid receptor A subunit α1 (GABA-A Rα1) in mouse brain cortex [22]. In PK^−^/^−^/TauVLW mice, treatment with a mixture of CBD and THC reduced the stress-related abnormal behaviors observed in vehicle-treated PK^−^/^−^/TauVLW mice, decreased the gliosis and deposition of tau and Aβ in the hippocampus and cerebral cortex, and increased the ratio of reduced/oxidized glutathione and autophagy [23]. In contrast to the putative effect of CBD on improving cognitive performance in preclinical AD models, the therapeutic potential of THC alone in the treatment of AD has not been widely documented. In the previous in vitro study using N2a-variant Aβ protein precursor cells, we observed that THC treatment inhibited the production and aggregation of Aβ40, the production of phosphorylated tau protein, decreased the intracellular phospho-GSK3β expression level, and enhanced mitochondrial function [24]. Two recent in vivo studies by Zimmer’s group demonstrated that treatment with 1 and 3 mg/kg/d of THC through osmotic minipumps for 28 consecutive days significantly improved the special learning performance of male C57BL6/J mice at the ages of 12 and 18 months [25,26] whereas a 1:1 combination of THC and CBD (1 mg/kg/d for each) failed to achieve the same effect as THC monotherapy [26]. Another study showed that single-dose intraperitoneal (IP) administration of THC at 0.002 mg/kg restored the cognitive function in 24-month-old female wild-type mice [27].

The present study focused on the evaluation of the modifying effect of low-dose THC treatment on spatial learning and memory and the associated mechanisms in aged 14~17 month-old APP/PS1 mice, which are known to exhibit elevated Aβ production and increased Aβ disposition in the brain with impaired memory function [28,29]. Results of this study provide a basis for the further development of low-dose THC as an alternative treatment for AD.

## 2. Results

### 2.1. No Cytotoxic Effect of THC and CBD Observed in Human PBMCs and Mouse N2a/APPswe

As shown in Figure 1, treatment with THC and CBD for 42 h over a concentration range of 0.0625–6.25 μM in human PBMCs and 0.0025–2.5 μM in mouse N2a/APPswe cells resulted in no significant changes in cell viability. This result indicates that THC and CBD had no cytotoxic effect on human PBMCs and mouse N2a/APPswe cells within the concentration range tested.

### 2.2. THC and CBD Reduced the Production of Aβ1–42 Peptide in N2a/APPswe Cells

The 40-residue peptide Aβ (Aβ1–40) represents the most abundant Aβ isoform in the brain [30] while the C terminally extended variant Aβ1–42, able to form insoluble fibril-like structures rapidly, is considered to be highly associated with AD pathology [31,32]. Early studies have demonstrated that CBD treatment significantly reduced Aβ1–40 production in vitro [14], while CBD–THC combination decreased soluble Aβ1–42 levels in vivo [21]. In this regard, treatment with CBD alone and in combination with THC were included as positive controls was the in vitro study to examine whether the inhibitory effect of THC alone on Aβ production in N2a/APPswe cells was comparable to that of CBD alone and CBD + THC. In this study, ELISA was used to examine the effect of THC and CBD treatment alone or in combination on the production of Aβ1–40 and Aβ1–42 in N2a/APPswe cells that constitutively produce high levels of Aβ protein due to the transfected mutant APP gene. As shown in Figure 2A, treatment with 10 nM THC, 100 nM THC, 100 nM CBD, and 100 nM of THC and CBD in combination for 24 h significantly decreased the Aβ1–40 production in N2a/APPswe cells by 23% (*p* < 0.05), 27% (*p* < 0.01), 21% (*p* < 0.05), and 32% (*p* < 0.001), respectively. The mean Aβ1–40 level in N2a/APPswe cells treated with 10 nM CBD was not significantly different from that of the vehicle control (Figure 2A.). Treatment with 10 nM THC, 100 nM THC, 10 nM CBD, 100 nM CBD, and 100 nM of THC and CBD in combination for 24 h significantly decreased the Aβ1–42 production in N2a/APPswe cells by 28% (*p* < 0.001), 35% (*p* < 0.001), 18% (*p* < 0.05), 19% (*p* < 0.05), and 31% (*p* < 0.001), respectively (Figure 2B). Moreover, the inhibitory effect of 100 nM THC on Aβ1–42 production was significantly greater than that of 10 nM CBD (*p* < 0.05. Figure 2). Treatment with THC and CBD alone or in combination for 42 h had no significant effect on the production of Aβ1–40 and Aβ1–42 in N2a/APPswe cells (*p* > 0.05 for all, Figure 2C,D) although treatments with 10 and 100 nM of THC for 42 h were able to decrease the Aβ1–42 production in N2a/APPswe cells by 16% and 24%, respectively (Figure 2D).

### 2.3. THC Treatment Improved Impaired Spatial Memory Performance of Aged APP/PS1 Mice

To evaluate the effect of low-dose THC exposure on the spatial reference memory of aged APP/PS1 transgenic mice, individual single-housed aged APP/PS1 transgenic mice were evaluated for the time taken to locate the escape platform (latency) and the number of errors using the RAWM test before and after vehicle or THC treatment and compared with the age-matched control of the non-transgenic (NTG) mice (Appendix A). To evaluate the effect of low-dose THC exposure on the spatial reference memory of aged APP/PS1 transgenic mice, individual single-housed aged APP/PS1 transgenic mice were evaluated for the time taken to locate the escape platform (latency) and the number of errors using the RAWM test before and after vehicle or THC treatment and compared with the age-matched control NTG mice. There was no significant effect of sex on either the latency or number of errors before the start of the treatments, suggesting the insignificance of effect modification by sex (Figure 3A,B). Therefore, the data from male and female mice were pooled for subsequent statistical analyses including the two-sample *t* test, one-way ANOVA with the post-hoc Bonferroni’s multiple comparisons test, and two-way ANOVA. Results of the two-way ANOVA for the baseline latency revealed a significant effect of time (*F*(2, 46) = 7.61, *p* = 0.0014) and group interaction (*F*(3, 23) = 7.24, *p* = 0.0014), while there was no significant effect for the time × group interaction (*F*(6, 46) = 0.759, *p* = 0.606). The post-hoc analysis indicated a significant difference in latency between the non-transgenic (NTG) control and individual transgenic groups, but no difference among the three transgenic groups in Trial 3 of the last block on Day 3 (Figure 3C). Results of two-way ANOVA for the baseline number of errors also showed a significant effect of time (*F*(2, 46) = 3.84, *p* = 0.0286) and group interaction (*F*(3, 23) = 9.65, *p* = 0.0003), but not for the time × group interaction (*F*(6, 64) = 0.393, *p* =0.88). The post hoc analysis indicated a significant difference in the number of errors between the NTG control and transgenic 0.02 mg/kg THC group in Trial 3 of the last block on day 2 and between the NTG control and transgenic 0.02 mg/kg and 0.2 mg/kg THC groups in Trial 3 of the last block on day 3. No difference was found among the three transgenic groups in Trial 3 of the last block on day 3 (Figure 3D). Taken together, it was demonstrated that the APP/PS1 mice displayed significant spatial memory deficits compared to the NTG mice before the THC treatment was initiated. The APP/PS1 mice were grouped based on the RAWM results so that there was no overt detectable difference in spatial memory among the three study groups.

Since the results of the RAWM test obtained after the 3-month vehicle or THC treatment indicated that the effect of sex on either the latency or number of errors was not significant, the data from male and female mice were pooled for subsequent statistical analyses (Figure 4A,B). Results of two-way ANOVA for the latency revealed a significant main effect of treatment (*F*(3, 23) = 10.51, *p* < 0.001) and time (*F*(4, 92) = 10.41, *p* < 0.001), but not for the time × treatment interaction (*F*(12, 92) = 1.14, *p* = 0.338). Results of two-way ANOVA for the number of errors showed a significant main effect of treatment (*F*(3, 23) = 16.74, *p* < 0.001) and the time × treatment interaction (*F*(12, 92) = 2.28, *p* = 0.014), but not for the time (*F*(4, 92) = 1.30, *p* = 0.277). Results of one-way ANOVA with post hoc Bonferroni’s multiple comparisons test demonstrated that the vehicle-treated aged APP/PS1 mice took a significantly longer time to locate the escape platform and made more errors than the aged NTG mice during the last three interval-separating blocks of training trials (*p* < 0.001. Figure 4C,D). In contrast, in Trial 5 Block 5, APP/PS1 mice treated with 0.02 and 0.2 mg/kg THC showed a significant decrease in the latency and number of errors compared with the vehicle group (*p* < 0.01 and *p* < 0.001 for 0.02 mg/kg and 0.2 mg/kg THC groups, respectively). No significant differences in the latency and number of errors were observed between the THC-treated APP/PS1 mice and aged NTG control mice (Figure 4C,D). These observations indicate that the spatial learning and memory of THC-treated APP/PS mice were superior to those of the vehicle-treated APP/PS1 mice and comparable to those of the NTG control mice. Overall, the superiority of THC-treated APP/PS1 mice over the control APP/PS1 mice was evident in that the control APP/PS1 mice consistently exhibited inferior acquisition with little improvement in the spatial memory throughout the RAWM test sessions, while the performance of THC-treated APP/PS1 mice improved markedly over the same training period. In addition, no significant difference in latencies and number of errors was found between the NTG control mice and THC-treated APP/PS1 mice (*p* > 0.05. Figure 4C,D), implicating that the memory deficits in aged APP/PS1 mice are reversed with THC treatment.

### 2.4. THC Treatment Resulted in Decreased Oligomeric Aβ Levels in Hippocampi of APP/PS1 Mice

It is well documented that the spatial learning and memory abilities and brain chemical levels of APP/PS1 mice are different from those of wild-type non-transgenic mice due to the increased secretion of Aβ peptides in the APP/PS1 mouse brain, which causes neurotoxicity and triggers memory impairment [33,34]. In this regard, we dissected the mice brains after the three-month THC treatment. Aβ plaques on the mouse hippocampus sections were examined using Congo Red staining. Results of the Congo Red staining showed that the Aβ plaque areas and number of Aβ plaques on the APP/PS1 mouse brain hippocampus sections were significantly higher than those on control NTG mouse brain hippocampus sections irrespective of treatment (Figure 5). Although treatment with 0.2 mg/kg of THC appeared to decrease the Aβ plaque areas and number of Aβ plaques by 28% and 27%, respectively, compared with the vehicle treatment, the difference was not statistically significant (Figure 5B,C). Nonetheless, the difference in Aβ plaque areas was statistically significant between the 0.02 mg/kg and 0.2 mg/kg THC treatment groups (*p* < 0.01, Figure 5B). The overall result of Congo Red staining of APP/PS1 mouse hippocampus sections revealed the limited effect of 3-month THC treatment on reducing the amyloid deposits in the hippocampi.

### 2.5. THC Treatment Reduced Oligomeric Aβ and Increased Monomeric Aβ Levels in Brain Tissues but Had No Significant Effect on Soluble and Insoluble Aβ Peptide Levels in Plasma

The levels of soluble and insoluble Aβ peptide in plasma were measured using ELISA (Figure 6A,B). The levels of Aβ monomer, Aβ oligomer, and total Aβ in brain tissues were determined using semi-quantitative western blotting analysis (Appendix A and Figure 6C–E). Treatment with THC at 0.02 and 0.2 mg/kg had no significant effect on the soluble and insoluble Aβ1–40 levels in plasma compared with the baseline levels before the start of the treatment (Figure 6A,B). In contrast, treatment with 0.2 mg/kg THC significantly increased the mean level of the Aβ monomer in the brain tissues of APP/PS mice by 167% compared with that in the control APP/PS1 brain tissues (*p* < 0.05. Figure 6C). The mean level of oligomeric Aβ in the cerebral cortex was significantly decreased by 34% (*p* < 0.05) and 61% (*p* < 0.01) in THC 0.02 and 0.2 mg/kg treatment groups, respectively, compared with the control TG group (Figure 6D). Although THC treatment tended to reduce the total Aβ levels in APP/PS1 brain tissues compared with the vehicle control, the difference in total Aβ levels was not significant among all APP/PS1 study groups (*p* > 0.05, Figure 6E). Nonetheless, the effect of THC on increasing the monomeric Aβ level (Pearson correlation coefficient r = 1.000, *p* < 0.001), reducing the oligomeric Aβ level (Pearson correlation coefficient r = −0.908, *p* < 0.001), and total Aβ level (Pearson correlation coefficient r = −0.906, *p* < 0.001) in brain tissues was in a dose-dependent manner (Figure 6C–E). The significant effect of THC treatment at 0.2 mg/kg on Aβ levels in the brain was also demonstrated by the fact that the total Aβ normalized Aβ monomer level in the 0.2 mg/kg THC group was significantly lower than those in the control and 0.02 mg/kg THC groups (*p* < 0.001 for both. Figure 6F), whereas the total Aβ normalized Aβ oligomer level in the 0.2 mg/kg THC group was significantly higher than those in the control (*p* < 0.001) and 0.02 mg/kg THC groups (*p* < 0.05. Figure 6G). Neither Aβ monomer nor Aβ oligomer nor total Aβ was detected in the brain tissues in control NTG mice by western blot analysis. Overall, results of the western blot analysis of Aβ levels in the brain tissues demonstrated that treatment with 0.2 mg/kg effectively prevented Aβ monomers from aggregating into Aβ oligomers.

### 2.6. THC Treatment Reduced Expression of Phospho-Tau and Total Tau and Decreased the Activity of GSK-3β through Increasing Expression of Phospho-GSK3β at Ser9 in APP/PS1 Brain Tissues

One of the hallmark pathologies that characterizes AD is neurofibrillary tangles (NFT), which are closely associated with hyperphosphorylated tau levels [7,8]. Dysregulation of GSK-3β is believed to be associated with the hyperphosphorylated tau levels [35,36]. The mitochondrial transcription factor A (TFAM) essential for genome maintenance, creatine kinase U-type (CKMT1) involved in energy metabolism, and mitochondrial fission factor (MFF) involved in the mitochondrial fission process are often decreased in neurodegenerative diseases including AD [37,38,39]. Results from the western blot analysis of brain tissue homogenates showed that the mean level of phospho-Tau was significantly lower in the control NTG brain tissues than those in the control TG (*p* < 0.05) and 0.02 mg/kg THC treatment (*p* < 0.01) groups (Figure 7A). Moreover, the mean expression levels of phospho-tau and total tau in the 0.2 mg/kg THC treatment group were significantly lower than those in the control TG (*p* < 0.01 for both phospho-tau and total tau) and 0.02 mg/kg THC (*p* < 0.01 for phospho-tau and *p* < 0.05 for total tau) treatment groups (Figure 7B). It was noted that the mean phospho-tau/tau ratios in the THC 0.02 and 0.2 mg/kg treatment groups were significantly higher than those in the control NTG (*p* < 0.01 for both THC treatment groups) and TG (*p* < 0.05 for both THC treatment groups) groups (Figure 7B), suggesting that the phospho-tau/tau ratio is not an appropriate marker for the efficacy of THC in the treatment of AD. As shown in Figure 7B, treatment with 0.2 mg/kg THC significantly increased the expression levels of phospho-GSK-3β at Ser9 in brain tissues compared with the NTG control, TG control, and THC 0.02 mg/kg treatments (*p* < 0.01 for all). THC treatment at 0.02 and 0.2 mg/kg significantly increased the total GSK-3β levels in brain tissues compared with the TG control treatment (*p* < 0.05 and *p* < 0.01 for 0.02 and 0.2 mg/kg THC treatment, respectively, Figure 7B). The mean total GSK-3β level in 0.2 mg/kg THC-treated APP/PS1 brain tissues was also significantly higher than that in the control NTG brain tissues (*p* < 0.05). However, it appeared that the phospho-GSK-3β/GSK-3β ratio was decreased by 31% and 26% in the 0.02 and 0.2 mg/kg THC treatment groups, respectively, compared with that in the control TG group, but the difference was not statistically significant (*p* > 0.05, Figure 7B). No significant difference in the TFAM, CKMT1, and MFF expression levels in the brain was found among all study groups (Figure 7B). Taken together, results of the western blot analysis demonstrated that THC treatment at 0.2 mg/kg significantly decreased phospho-tau and total tau levels and increased the phospho-GSK-3β (Ser9) and total GSK-3β levels but had no effect on the mitochondrial function and energy metabolism in the brain.

### 2.7. THC Treatment Had Little Effect on Neuropathologic Changes in the Hippocampi of APP/PS1 Mice

The immunohistochemistry was employed to assess the effect of THC treatment on AD neuropathologic change through the evaluation of the protein expression of phospho-tau, ionized calcium-binding adaptor molecule (Iba1), neuronal nuclei (NeuN), CB1, and GSK-3β in hippocampi. As shown in Figure 8, the areas positively stained for phospho-tau and Iba1 on the APP/PS1 mouse hippocampus tissue sections were significantly greater than those on the control NTG mouse hippocampus tissue sections regardless of treatment. Although treatment with 0.2 mg/kg THC appeared to reduce the mean phospho-tau-positive area by 21% compared with the control TG group, the difference was not statistically significant (Figure 8). The areas positively stained for NeuN on the hippocampus tissue sections of the control TG and 0.02 mg/kg THC groups were significantly greater than that of the control NTG group (*p* < 0.05 for both, Figure 8), while no significant difference in the NeuN-positive areas was found between the control NTG and 0.2 mg/kg THC groups (Figure 8). No significant difference in the percent areas stained positively for CB1 and GSK-3β in the hippocampi was found among all study groups (Figure 8). Taken together, our data suggest that THC treatment has little effect on reversing the neuropathologic change of AD.

### 2.8. THC Treatment Had Little Effect on Cytokine Levels in Plasma

Given the evidence that plasma cytokine levels are altered in AD and correlated with the disease progress [40], the effect of THC on plasma cytokine level was evaluated after the completion of the 3-month THC treatment at 0.02 and 0.2 mg/kg. Results of the cytokine plasma levels determined by ELISA showed that THC tended to decrease the IL2 and IL10 plasma levels in a dose-dependent manner (Pearson correlation coefficient r = −0.902 and −0.793, respectively). However, the difference between the APP/PS1 control group and either of the THC treatment groups was not statistically significant (*p* > 0.05 for all, Figure 9). THC treatment at 0.02 and 0.2 mg/kg also had little effect on all of the other cytokines including the IL4, IL6, IL12, and IL17 plasma levels compared with the vehicle treatment in either non-transgenic or APP/PS1 mice, suggesting that THC treatment at 0.02 and 0.2 mg/kg is unlikely to induce any immunotoxicity (Figure 9).

## 3. Discussion

Extensive preclinical and clinical studies have focused on the ability of CBD to prevent, slow, or stop the cognitive damaging effect of AD [12,13,14,15,16,17,18,19,20]. In contrast, relatively few studies have examined the therapeutic potential of THC in AD. The general perception is that THC intoxication impairs cognitive functions ranging from fundamental motor skills such as walking and grasping to higher-order executive functioning such as working memory, inhibitory control, cognitive flexibility, and response planning [41]. In mice, THC doses inducing acute intoxication that increased spontaneous anxiety-like and locomotor behavior were found to be 1 mg/kg for intravenous administration [42] and 10 mg/kg for intraperitoneal administration [43]. In this regard, the THC doses used in this study were much lower than the dose that could induce the anxiety-like behavior in mice. Contrary to common belief, a handful of preclinical studies have revealed the beneficial effect of in vivo THC treatment at doses ranging of 0.002 and 3 mg/kg against the age-related decline in cognitive performance of aged mice [25,26,27]. Unlike the documented in vivo studies [25,26,27], the present study was carried out on aged APP/PS1 mice, and the animals were treated with 0.02 and 0.2 mg/kg of THC once every other day, which were lower than the doses of 1 and 3 mg/kg/d reported by Zimmer’s group [25,26]. Results of our study confirmed that aged APP/PS1 mice exhibited significantly elevated Aβ production and impaired spatial memory that recapitulated the early-stage AD in humans (Figure 3, Figure 4 and Figure 5 and Figure 6C–G), suggesting that aged APP/PS1 mice are a better model than aged wild-type mice for evaluating the potential of THC in the treatment of AD.

In this study, the results of the RAWM test demonstrated that aged APP/PS1 mice treated with THC at the dose of 0.02 and 0.2 mg/kg once every other day for three months exhibited a significantly reduced number of errors and decreased latency values compared with the control APP/PS1 mice. The observed ability of low-dose THC treatment to improve the spatial memory performance of aged APP/PS1 mice coincided with its inhibitory effect on Aβ production and aggregation, implicating that THC might reduce Aβ toxicity by targeting its polymerization in the brain. Results of the in vitro study showed that treatment of N2a/AβPPswe cells with THC at non-cytotoxic concentrations of 10 and 100 nM for 24 h significantly reduced the cells’ production and secretion of Aβ (Figure 2A,B). Consistent with the in vitro observation, the result of the Congo Red staining of the hippocampus tissue sections demonstrated that treatment with 0.2 mg/kg THC tended to reduce the amyloid deposits in hippocampi. However, the difference in the mean area positively stained for Congo Red and number of amyloid deposits between the control TG and 0.2 mg/kg THC treatment groups was not statistically significant (Figure 5). Based on this finding, it might be possible that an extended THC treatment period would result in a significant reduction in the amyloid deposits in the brain. Nonetheless, treatment with 0.2 mg/kg THC significantly increased the Aβ monomer level and decreased the Aβ oligomer level in APP/PS1 mouse brain homogenates (Figure 6). Given the overall observations in this study as well as the results of our previous thioflavin T fluorescence assay showing the inhibitory effect of THC on Aβ1–40 aggregation in vitro [24], it is speculated that low-dose THC treatment has the potential to reduce amyloid deposits in the brain by preventing the aggregation of Aβ monomers and the formation of Aβ oligomers. Further studies are warranted to investigate the effect of THC on the polymerization process of Aβ through direct monitoring of the Aβ monomer to oligomer transition in vivo using fiber-based fluorescence correlation stereoscopy [44].

Aβ and tau are the two principal toxic species in AD, which constitute the core building blocks for extracellular amyloid plaques and intraneuronal neurofibrillary tangles, respectively [45,46,47]. Accumulating evidence suggests that Aβ functions upstream of tau in AD pathogenesis, driving tau pathology through the modulation of protein kinases and phosphatases that regulate tau phosphorylation and the induction of tau misfolding, while tau in turn mediates the synaptic toxicity of Aβ at the postsynaptic compartment and dendritic spines [46,47,48]. In this study, we observed that treatment of THC at 0.2 mg/kg significantly decreased not only the Aβ oligomer level but also the phospho-tau and total tau levels in APP/PS mouse brain tissues (Figure 6D and Figure 7). This finding is in line with our previous observation that THC treatment inhibited Aβ aggregation in vitro and suppressed phosphorylated Tau production in cultured N2a/APPswe cells [24]. One possibility would be that THC treatment interrupts the pathological feedback loop between Aβ and tau. Another possibility is that THC targets Aβ and tau through different molecular mechanisms. Further studies are warranted to pinpoint the mechanism underlying the dual effect of low-dose THC treatment on Aβ and tau.

An intriguing finding in this study was that 3-month treatment with 0.2 mg/kg THC significantly increased the expression level of phospho-GSK-3β at Ser9 and total GSK-3β in the brain tissues of APP/PS1 mice compared with the vehicle control treatment in NTG and TG mice (Figure 7). The activity of GSK-3β is regulated by phosphorylation. Phosphorylation of GSK-3β at the N-terminal Ser9 inhibits the activity of GSK-3β, while phosphorylation at Tyr216/Tyr279 increases its activity [49,50]. Activation of GSK-3β is regulated by the insulin and Wnt pathways [51]. Aβ oligomers can bind to the Frizzled family of Wnt receptors, preventing Wnt from downregulating GSK-3β [52], or act as an antagonist of the insulin receptor, augmenting GSK-3β activity [53]. Moreover, chronic exposure of mice to Aβ has been shown to downregulate Akt phosphorylation, which in turn provokes GSK-3β activation [54]. The activated GSK-3β may then phosphorylate tau protein and promote the formation of neurofibrillary tangles [55,56]. In this regard, GSK-3β plays a pivotal role in linking Aβ toxicity and tau hyperphosphorylation [55,57]. The observation that THC inhibited GSK-3β activity by increasing the phosphorylation of GSK-3β at Ser9 was similar to other studies in which lithium chloride, a mood stabilizing agent, inhibited GSK-3β activity by increasing Ser9 phosphorylation [58,59]. Taken together, it is speculated that low-dose THC treatment suppresses the activity of GSK-3β not only by increasing the phosphorylation of GSK-3β at Ser9 but also through its inhibitory effect on the formation of Aβ oligomers that would otherwise prevent insulin receptor and Frizzled from downregulating GSK-3β activity (Figure 10). The decreased GSK-3β activity also contributes to the decreased tau phosphorylation.

Our previous in vitro study showed that THC treatment increased the rate of mitochondrial oxygen consumption [24]. Therefore, we evaluated the effect of in vivo THC treatment on modulating the brain expression of TFAM, CKMT1, and MFF proteins, which are prominent effectors involved in genome maintenance, energy metabolism, and mitochondrial fission process [37,38,39]. TFAM controls the transmission and expression of mitochondrial DNA (mtDNA), packaging mtDNA into mitochondrial nucleoids that encode essential subunits of the mitochondrial oxidative phosphorylation system [60]. CKMT1 is expressed predominantly in neurons and its downregulation implicates neuronal cell loss [61,62]. MFF activity is required for the maintenance of axonal mitochondrial size [63], and loss of MFF results in increased neuronal cell loss, astrogliosis, and neuroinflammation, suggesting that MFF plays an important role in maintaining the delicate balance between mitochondrial fusion and fission [64]. Although results of the western blot analysis of the brain homogenate samples showed that the expression of TFAM and MFF proteins in the THC 0.2 mg/kg group was increased by 20% and 30%, respectively, compared with those in the control TG group, the difference was not statistically significant (Figure 7). This result suggests that low-dose THC treatment has no significant impact on the brain mitochondrial respiratory chain dysfunction that could lead to cannabis-related stroke [65].

In addition, THC treatment at 0.2 mg/kg appeared to decrease concentrations of IL2, IL10, and IL12 in plasma compared with the vehicle treatment in APP/PS1 mice although the difference was not statistically significant. Since cytokine concentrations in plasma may not reflect cytokine levels or inflammatory events in the brain [66], further study is needed to evaluate the effect of THC on the cytokine levels in the hippocampi. In addition, results of the IHC analysis showed that the % area stained positively for Neu in the control TG and 0.02 THC treatment groups was significantly higher than that in the control NTG group (Figure 8). Since one mouse strain may have more abundant NeuN-positive cells than another mouse strain [67], it is speculated that the observed significant difference in the NeuN-positive area between NTG and APP/PS1 mice is mouse strain-related.

In summary, the results of this study demonstrated that IP administration of 0.02 and 0.2 mg/kg THC was safe and effective in improving the spatial memory performance of aged APP/PS1 mice that exhibited significant memory decline compared with the aged NTG mice. The memory-improving effect of low-dose THC treatment is associated with its inhibitory effect on Aβ aggregation, GSK-3β activity, and tau phosphorylation in the brain (Figure 10).

## 4. Materials and Methods

### 4.1. Materials, Cell Line, and Animals

CBD (CAS No. 13956-29-1, lot CBD021615-01) and THC (CAS No. 1972-08-3, lot D9101515-01) were manufactured by Austin Pharma, LLC (Round Rock, TX, USA). The CBD and THC stock solutions with a concentration of 5 mg/mL (15.9 mM) and 0.5 mg/mL (1.59 mM), respectively, were prepared in ethanol and stored in a −20 °C freezer and secured with two locks. Lyophilized Aβ1-42 peptide (Catalog No.: 1409-rPEP-02. Biomer Technology, Pleasanton, CA, USA) was suspended in pre-chilled 1,1,1,3,3,3-hexafluoro-2-popanol (HFIP) on ice to make 1 mM of the Aβ solution with a concentration of 1 mM. The Aβ solution was kept on a rotator at RT for 24 h until completely dissolved. The Aβ solution was then aliquoted in 10 µL into pre-chilled tubes and centrifuged at 1000× *g* using a Speedvac vacuum concentrator (Thermo Fisher Scientific, Waltham, MA, USA) to evaporate HFIP. The dried solutes were stored at −80 °C. Before use, Aβ was reconstituted in 1% NH_4_OH to 10 mg/mL and diluted into a working solution with 1x Tris buffered saline (TBS). All other chemicals and solvents were obtained from commercial sources.

N2a (ATCC^®^ CCL-131TM) mouse brain neuroblasts and primary human peripheral blood mononuclear cells (PBMC) (ATCC^®^ PCS-800-011TM) were purchased from the American Type Culture Collection (ATCC; Manassas, VA, USA) and propagated in modified Dulbecco’s modified Eagle’s (DMEM) medium (Catalog No.: 10-013-CV. Thermo Fisher Scientific, Waltham, MA, USA) containing 10% heat-inactivated fetal bovine serum (FBS) (GibcoTM#10082147. Thermo Fisher Scientific, Waltham, MA, USA), 100 U/mL penicillin, and 100 μg/mL streptomycin. N2a cells stably expressing human APP carrying the K670N/M671L Swedish mutation (N2A/APPswe) were grown in Dulbecco’s modified Eagle medium (DMEM) containing 10% fetal bovine serum, 100 U/mL penicillin, 100 mcg/mL streptomycin, and 400 mcg/mL G418 (Thermo Fisher Scientific, Waltham, MA, USA), at 37 °C in the presence of 5% CO_2_.

The APP/PS1 transgenic mice used in this study were a unique mouse strain that can be challenging to breed. They are a double transgenic mouse line expressing Amyloid Precursor Protein and Presenilin 1 (APP/PS1) and generated by crossing Tg2576 APP mice [68] with line 5.1 PS1 mice [69]. The Tg2576 APP mice were derived from a C57B6/SJL × C57B6 background, while the line 5.1 PS1 mice were derived from a Swiss Webster/B6D2F1 × B6D2F1 background. The double transgenic APP/PS1 progeny possesses a genetic background that is a mixture of the backgrounds of Tg2576 APP mice and line 5.1 PS1 mice. The resultant APP/PS1 mouse strain possesses transgene-positive APP_K670N, M671L_ and PS1_M146L_, which is rare in that only one fourth of the newborn population in each generation may possess all three mutations. Individual animals were genotyped after weaning and group-housed with free access to water and rodent chow [34]. The APP/PS1 transgenic mice and non-transgenic control mice were maintained at the University of South Florida Department of Comparative Medicine pathogen free animal facility. All mice were acclimated in our vivarium at 21–24 °C at 40–60% humidity with a 12-h light/dark cycle. Mice used in the radial arm water maze test were housed individually to avoid disturbing one another, which has been a standard protocol used by our colleagues [33] as well as by other research groups [70,71].

All animal experiments were approved by the Institutional Animal Care and Use Committee (IACUC) (Project ID: IS00000959. Approved on 4 January 2018) and performed according to the National Institute of Health (NIH) guidelines.

### 4.2. In Vitro Cytotoxicity Assay

The MTT (3-[4,5-dimethylthiazole-2-yl]-2,5-diphenyl-tetrazolium bromide) assay was performed to evaluate the cytotoxic effect of THC and CBD on N2a/AβPPswe cells and human PBMC cells. In brief, N2a/AβPPswe cells and human PBMC cells were seeded in 96-well tissue culture plates at 12,000 cells/well and 125,000 cells/well, respectively, and allowed to attach overnight. The next day, culture media containing either THC or CBD at various concentrations were added to appropriate wells. After the cells were treated for 40 h, an aliquot of 20 µL of 5 mg/mL MTT reagent was added to each well and incubated for 3.5 h at 37 °C. Then, all of the media was carefully removed, and 150 µL of MTT detection reagent was added to each well. The culture plate was covered with aluminum foil and agitated on an orbital shaker for 15 min. Final OD values were read at 575 nm with a reference filter of 620 nm on a Bio-Tek Synergy HT plate reader (Winooski, VT, USA). Cell viability was calculated as: Cell viability = (OD sample−OD blank)/(OD control—OD blank) ×100%.

### 4.3. Determination of Aβ Level Using Sandwich Enzyme Linked Immunosorbent Assay (ELISA)

To evaluate the effect of THC and CBD on Aβ production, N2a/APPswe cells were treated with (1) 100 nM THC; (2) 10 nM THC; (3) 100 nM CBD; (4) 10 nM CBD; or (5) 100 nM THC and 100 nM CBD in combination for 24 or 42 h. The cell culture supernatant samples were subjected to ELISA for the determination of Aβ concentrations. The sandwich ELISA assay (MegaNanoDiagmostics Inc., Tampa, FL, USA) was carried out by using the goat anti-N-terminal Aβ antibody (50 µL, 10 µg/mL) as the capture antibody to coat a 96-well plate (Immuion 4HBX). The antibody-coated plate was incubated overnight at 4 °C. After the coating, the solution was removed, the plate was washed and then blocked by 300 µL of blocking buffer (1X PBS containing 2% Bovine albumin), incubated for 1 h at room temperature, and then washed four times with 1X PBS. Six Aβ standard solutions (1000, 500, 250, 125, 62.5, and 31.25 pg/mL) were prepared by serial dilution. Tissue culture supernatant samples were collected from treated cells and diluted with diluent buffer containing the protease inhibitor (1:10, *v*/*v*). Aliquots of 100 μL of individual samples or standards were added to respective wells in triplicate and incubated with 50 µL of detection anti-Aβ antibody for 2 h at room temperature. After the plate was washed four times with 1X PBS, 100 μL of secondary antibody (anti-rabbit-HRP) was added to each well and incubated at 37 °C for 45 min on a shaker. After the final wash step, 100 μL of 3,3′,5,5′-tetramethylbenzidine (TMB) substrate was added and incubated for 10–30 min in the dark. The reaction was halted by adding 100 μL stop solution for detection at 450 nm using a Bio-Tek Synergy HT plate reader (Winooski, VT, USA). A 4-parameter regression was used to generate the standard curve. The Aβ levels in individual samples were expressed as the percent change over the average Aβ level in the vehicle-treated control samples.

### 4.4. In Vivo Treatment Protocol

Non-transgenic control mice and APP/PS1 transgenic mice (14 months of age) were divided into four groups: (1) non-transgenic control; (2) APP/PS1 transgenic control; (3) APP/PS1—0.02 mg/kg THC treatment; and (4) APP/PS1—0.2 mg/kg THC treatment. Prior to the start of any treatment, groups were constituted according to the following criteria in order to minimize between-group variability. Groups were balanced with respect to gender and plasma Aβ level, body weight, and baseline memory test results within gender. Individual animals were given an intraperitoneal (IP) injection of PBS or THC once every other day for three months. Individual animals were subjected to the radial arm water maze test before the treatment started and after the 3-month treatment ended. All animals were euthanized at the end of the experiment. Plasma samples prepared from whole blood collected were stored at −80 °C before being subjected to the determination of THC concentrations using liquid chromatography tandem mass spectrometry and the levels of selected inflammation or anti-inflammation related cytokines. Brains were excised and halved. The left cerebral hemispheres were fixed in 4% PFA before being subjected to immunochemistry.

### 4.5. Radial Arm Water Maze

Behavioral assessment for spatial learning and memory was carried out with aged non-genetic (NTG) and APP/PS1 transgenic (TG) mice using the radial arm water maze (RAWM) method with minor modifications [72,73,74]. The apparatus was a 6-arm maze in which one arm served as the start arm and the other five arms had a submerged escape platform at the end of each arm [72,75]. In the first trial, the mouse was allowed to enter any arm to find an escape platform. The animal was then kept in a holding cage for a specified length of time while the platform it found was removed. In the second trial, if the mouse remembered the arm that it chose before, it should not enter that arm again but choose another one. The process continued until the animal found all platforms [75]. The learning and memory performance of each aged mouse were assessed based on the time spent on locating the escape platform (i.e., latency) and the number of errors measured in a one-minute time frame. Before the start of the vehicle or THC treatment, individual 14-month-old NTG and APP/PS mice were randomly divided into cohorts of four, which were mixed with respect to sex and/or genotype and then subjected to a 3-day RAWM protocol and underwent 15 trials per day. Individual mice were allowed up to 60 s to swim through the maze and locate the escape platform. Their escape latency and number of errors were recorded. In brief, on day 1, 15 trials were run in five blocks of three. The cohorts of mice were exposed sequentially to the running. The first cohort of mice was not exposed to the second block until the last cohort completed the running, allowing an extended rest period for individual cohorts [74]. On day 2, the same cohorts of mice were tested in the same manner as that on day 1, except this time using the hidden platform for all trials. On day 3, cohorts of mice were subjected to the reversal trials with a new goal arm location for each mouse not adjacent to the initial goal arm and the hidden platform used for all trials [76]. Based on the results of latency and number of errors obtained in Trial 3 of the last block on day 3, APP/PS1 mice were randomly divided into the vehicle control, 0.02 mg/kg THC, and 0.2 mg/kg THC groups.

Since the older mice may become exhausted from swimming after undergoing 15 trials per day without much rest, the aged NTG and APP/PS1 mice were subjected to a 15-day RAWM protocol after they were treated with either the vehicle or THC for three months [76,77]. In brief, individual mice completed five trials per day (inter-trial interval ~10 min) for 15 consecutive days. At the start of the first trial, the mouse was released from the start arm and allowed up to 60 s to swim through the maze and locate the submerged escape platform. After the platform was found, the mouse remained on it for 20 s before being returned to a heated holding cage. For the second trial, the mouse was placed back in the start arm and allowed to locate another escape platform. The process repeated until the animal found all platforms. Any mouse that was unable to find the platform after 60 s was guided to the nearest platform. The number of errors and latency were recorded in the same way as the pre-treatment evaluation.

### 4.6. Protein and Peptide Extraction

Proteins from cell lysis or brain tissue samples were extracted with RIPA lysis and extraction buffer (Catalog No.: 89900. Thermo Fisher Scientific, Waltham, MA, USA) and quantified with the BCA Protein Assay Kit (Catalog No. A53225. Thermo Fisher Scientific, Waltham, MA, USA) according to the manufacturer’s protocols. The soluble and insoluble Aβ peptide extraction was based on the protocol by Izco M, et al. with a slight modification [78]. A total of 6 M of guanidine-HCl instead of formic acid was used to dissolve the insoluble Aβ pellet.

### 4.7. Semi-Quantitative Western Blotting Analysis of Brain Tissue Homogenate Samples

Brain tissue homogenates were prepared from cerebral cortex samples obtained from individual APP/PS1 mice treated with the vehicle or THC for three months and the control NTG mice. Brain tissue homogenate samples were reduced and denatured at 72 °C for 10 min before being subjected to the Tris-Tricine SDS-PAGE as previously described [24]. Immunoblotting was carried out with the following primary antibodies (1:1000 for each): glycogen synthase kinases 3β (GSK-3β) (12456, Cell Signaling Technology, Danvers, MA, USA), phospho-GSK-3β at Ser9 (5558, Cell Signaling Technology), total tau (ab76128, Abcam), phospho-tau (ab92676, Abcam), TFAM (ab138351, Abcam), CKMT1 (ab198257, Abcam), and MFF (ab129075, Abcam). Blots were incubated with horseradish peroxidase–conjugated secondary antibodies (1:5,000; Thermo Fisher Scientific, Waltham, MA, USA) and immunoreactive protein bands were visualized by the enhanced chemiluminescence system (PerkinElmer, Waltham, MA, USA). The membrane was then stripped and re-probed with β-actin (3700, Cell Signaling Technology, Danvers, MA, USA) as a loading control. Band areas of individual proteins were quantified using ImageJ software version 1.52a (https://imagej.nih.gov/ij/ (accessed on 30 December 2021)). Normalization for loading differences was achieved by dividing the densitometry values for individual proteins by the densitometry values for β-actin in the same lane. Protein expression levels in the control NTG and drug-treated samples were expressed as relative to those in the control TG samples.

### 4.8. Alkaline Congo Red Staining

Identification of Aβ plaques in brain sections was carried out using the alkaline Congo Red staining method as described elsewhere [79]. In brief, tissue sections were mounted on slides and left to dry overnight. Two liters of stock alcoholic saturated sodium chloride was prepared (NaCl in 80% ethanol), and the Congo Red solution (2 g Congo Red in 1 L alcoholic saturated sodium chloride) was stirred overnight at room temperature. The sections were hydrated for 30 s in distilled water then incubated with alkaline sodium chloride for 20–30 min (2 mL 1 M NaOH was added to 198 mL saturated alcoholic sodium chloride immediately before use). After being stained in an alkaline Congo Red solution (2 mL 1 M NaOH was added to 198 mL filtered Congo Red solution within 15 min prior to use) for 30 min, the slides were dehydrated through a series of alcohol solutions of increasing alcoholic concentration, cleared in xylene, and mounted under a coverslip.

### 4.9. Immunohistochemistry (IHC)

The left cerebral hemispheres were fixed in 4% PFA, dehydrated through a series of sucrose solutions, and sectioned at 25 µm thickness. The brain sections were incubated overnight at 4 °C with the primary antibodies (1:100 dilution for all) specific to the protein of interest including Aβ (developed in-house), total tau (ab76128, Abcam, Cambridge, MA, USA), phospho-tau (ab92676, Abcam, Cambridge, MA, USA), GSK3β (12456, Cell Signaling Technology, Danvers, MA, USA), Iba-1 (ab178846, Abcam, Cambridge, MA, USA), neuronal nuclei (NeuN) (ab177487, Abcam, Cambridge, MA, USA), and CB1 (ab3558, Abcam, Cambridge, MA, USA). After incubation with the primary antibody, the brain sections were subjected to a 1-h incubation with the biotinylated secondary antibody according to the manufacturer’s protocol (Vector Laboratories). Light microscopy staining was achieved with the standard biotin-streptavidin/HRP procedure and DAB chromogen as described elsewhere [80]. The sections were then counterstained with hematoxylin and mounted under coverslips. For each protein of interest, three sections were selected from the same hippocampus layer of the brain and used for analysis. All three measurements were averaged for each mouse to yield the value for further statistical analysis. The abnormal overstained or cracks were excluded from the analysis field.

### 4.10. Measurement of Plasma Cytokine Levels Using Multiplex Assay

Cytokine expression levels in plasma were determined using the Milliplex MAP mouse cytokine magnetic bead panel (Catalog #: MCYTOMAG-70K, MilliporeSigma, Burlington, MA, USA) according to the manufacturer’s protocols. In brief, plasma samples and standards were diluted with sample dilution buffer and mixed with magnetic beads and then the detection antibody. Immediately after the wash, the plate was read on a Bio-Plex^®^ MAGPIX^TM^ Multiplex Reader (Bio-Rad Laboratories, Inc. Hercules, CA, USA). The concentration of each cytokine in each sample was calculated using the standard curve of each cytokine.

### 4.11. Statistical Analyses

Statistical analyses were performed using Prism Graph Pad 5.0 (Graph Pad, San Diego, CA, USA). Data were presented as the mean ± standard deviation (SD) unless otherwise indicated. A paired sample *t* test was used to compare two means from the same animal. Comparison of means between two independent groups was conducted using the two-sample *t* test. Comparison of means between more than two independent groups was carried out using one-way ANOVA followed by the post hoc Tukey–Kramer multiple comparison test or Bonferroni’s multiple comparisons test when applicable. Results of latency to reach the platform and number of errors obtained from the RAWM test were analyzed using the two-way repeated measure ANOVA with time and treatment as factors. Pearson’s correlation coefficient was used to describe the strength of the linear correlation between two variables. A two-sided P-value of less than 0.05 was considered statistically significant.

## 5. Conclusions

Treatment with low-dose THC at 0.2 and 0.02 mg/kg improved the spatial learning of aged APP/PS1 mice by reducing the expression levels of oligomeric Aβ, phospho-tau and total tau and decreasing the activity of GSK-3β without eliciting any psychotropic or immunomodulatory effects, suggesting that low-dose THC is a safe and effective treatment for AD.

## Figures and Tables

**Figure 1 ijms-23-02757-f001:**
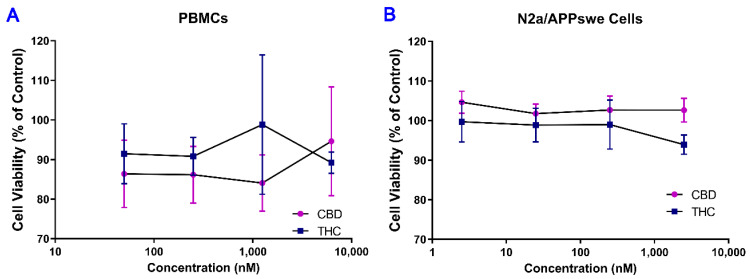
Determination of cytotoxicity of THC and CBD in (**A**) PBMCs and (**B**) N2a/APPswe cells. Concentration-effect curves demonstrated that treatment with THC or CBD for 40 h had no significant effect on the viability of cultured PBMCs and N2a/APPswe cells over a concentration range of 0.0625–6.25 μM and 0.0025–2.5 μM, respectively. Data are expressed as mean ± standard deviation (SD) (*N* = 3 for PBMCs. *N* = 5 for N2a/APPswe cells). Error bars denote SD.

**Figure 2 ijms-23-02757-f002:**
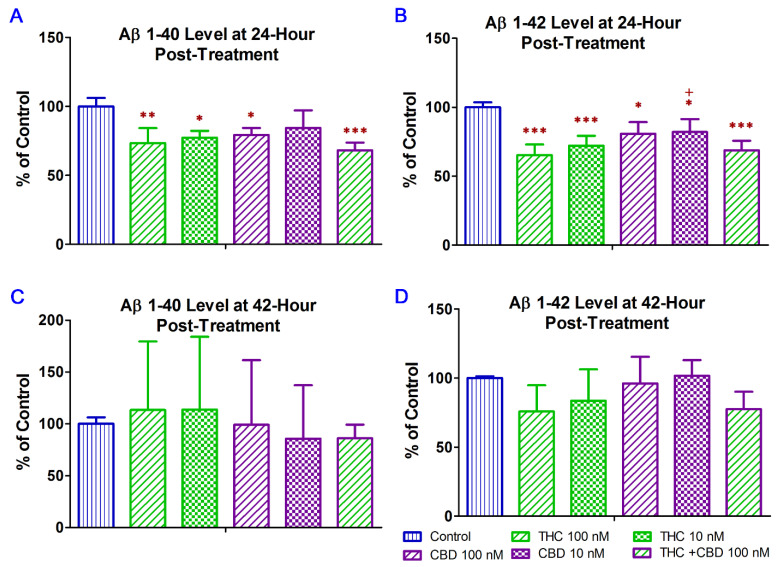
Effect of THC and CBD treatment alone or in combination on Aβ1–40 and Aβ1–42 production in N2a/APPswe cells at 24 (**A**) for Aβ1–40; (**B**) for Aβ1–42) and 42 (**C**) for Aβ1–40; (**D**) for Aβ1–42) hours after the treatment. The production of Aβ1–40 and Aβ 1–42 in the cell culture supernatant was determined by ELISA. After N2a/APPswe cells were treated with THC and CBD alone or in combination for 24 h, Aβ1–40 and Aβ 1–42 levels in the supernatant reduced significantly by 16~32% and 18~35%, respectively, compared with those in the non-treated control samples. No significant changes in the Aβ1–40 and Aβ 1–42 levels in the supernatant were found between the non-treated and treated samples after the 42-h treatment. Data are expressed as mean ± SD (*N* = 4). SD is denoted by the error bars. * *p* < 0.05, ** *p* < 0.01, and *** *p* < 0.001 compared with the control group and + *p* < 0.05 compared with the 100 nM THC treatment group using one-way ANOVA followed by the Tukey–Kramer post hoc multiple comparison test.

**Figure 3 ijms-23-02757-f003:**
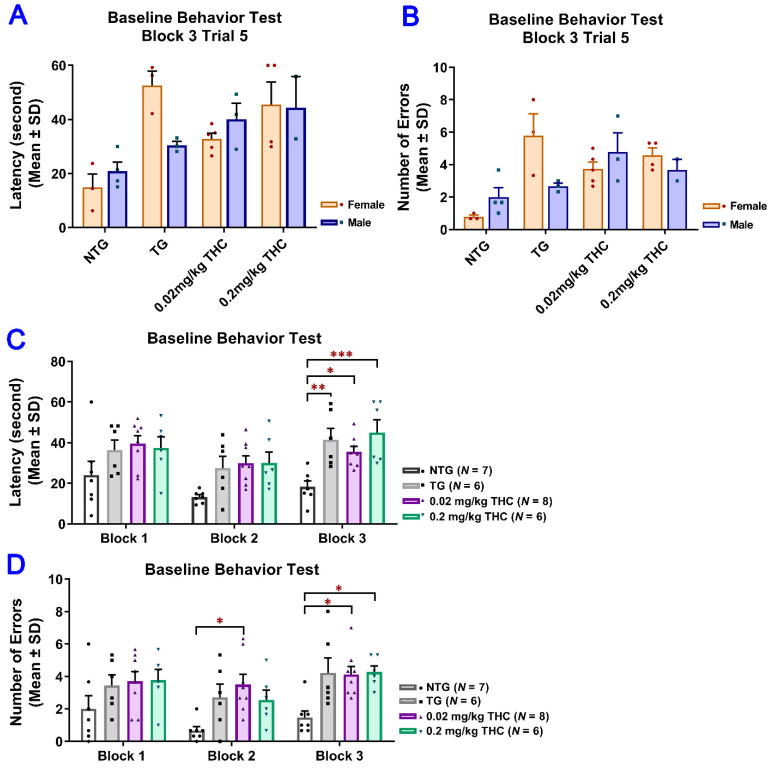
Evaluation of the baseline spatial reference memory of 14-month-old APP/PS1 mice in three study groups (i.e., the control transgenic control (TG), 0.02 mg/kg and 0.2 mg/kg THC treatment groups as well as 14-month-old non-transgenic (NTG) mice using the radial arm water maze (RAWM) test. Individual mice were subjected to five blocks of trials each day for three days with each block containing three trials. (**A**) No significant difference in baseline latency between male and female mice within individual study groups (*p* = 0.56, *p* = 0.06, *p* = 0.51 and *p* = 0.947 for NTG, TG, 0.02 mg/kg and 0.2 mg/kg THC groups, respectively, using the Independent sample *t* test). (**B**) No significant difference in baseline number of errors between male and female mice within individual study groups (*p* = 0.356, *p* = 0.299, *p* = 0.513, and *p* = 0.513 for NTG, TG, 0.02 mg/kg, and 0.2 mg/kg THC groups, respectively, using the Independent sample *t* test). (**C**) Significant increase in the baseline latency in TG (*p* < 0.01), 0.02 mg/kg THC (*p* < 0.05), and 0.2 mg/kg THC (*p* < 0.001) groups compared with the NTG control group in Trial 3 of the last block on day 3. (**D**) Significant increase in the baseline latency in TG (*p* < 0.01), 0.02 mg/kg THC (*p* < 0.05), and 0.2 mg/kg THC (*p* < 0.001) groups compared with the NTG control group in Trial 3 of the last block on day 3. Data are expressed as mean ± SD. SD is denoted by the error bars. A comparison of mean latency and number of errors between female and male animals in individual study groups was made with multiple *t*-tests with correction for multiple comparisons using the Holm–Sidak method. Comparison of mean baseline latency and number of errors among different study groups were conducted using one-way ANOVA with the post hoc Bonferroni’s multiple comparisons test. * *p* < 0.05, ** *p* < 0.01, and *** *p* < 0.001 compared between the control NTG (*N* = 7), control TG (*N* = 6), 0.02 mg/kg THC (*N* = 8), and 0.2 mg/kg THC (*N* = 6) groups.

**Figure 4 ijms-23-02757-f004:**
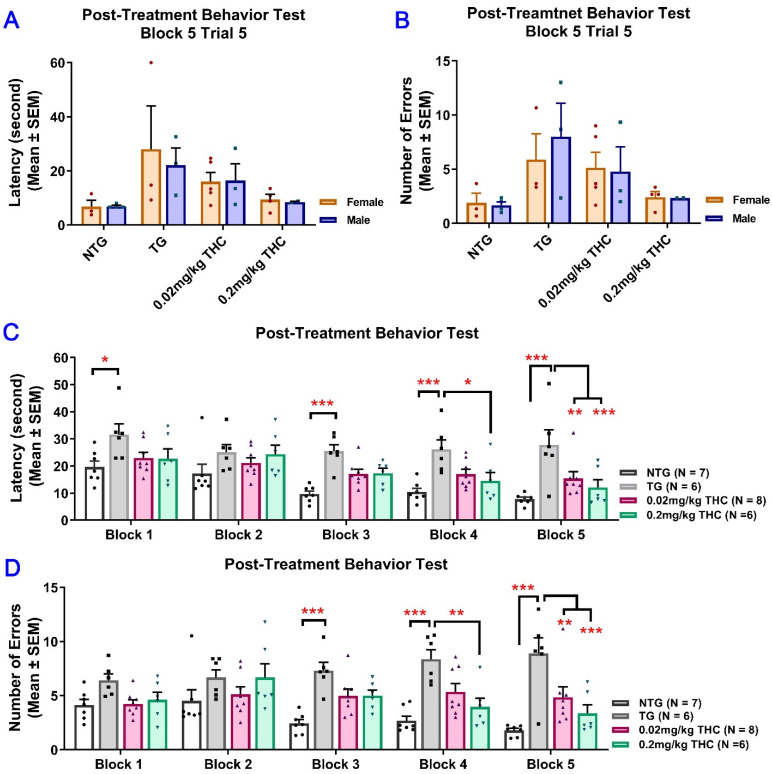
Evaluation of the effect of THC treatment on improving the spatial reference memory of 14-month-old APP/PS1 mice using the RAWM test. Individual mice were subjected to five trials per day for 15 consecutive days with each block containing 15 trials. (**A**) No significant difference in latency between male and female mice within individual study groups in Trial 5 of Block 5 (*p* = 0.998, *p* = 0.996, *p* = 0.998, and *p* = 0.996 for NTG, TG, 0.02 mg/kg, and 0.2 mg/kg THC groups, respectively, using the Independent sample *t* test). (**B**) No significant difference in number of errors between male and female mice within individual study groups in Trial 5 of Block 5 (*p* = 0.992, *p* = 0.979, *p* = 0.992, and *p* = 0.992 for NTG, TG, 0.02 mg/kg, and 0.2 mg/kg THC groups, respectively, using the Independent sample *t* test). (**C**) Significant decrease in the latency in NTG control (*p* < 0.001), 0.02 mg/kg THC (*p* < 0.01, and 0.2 mg/kg THC (*p* < 0.001) groups compared with the TG control group in Block 5 Trial 5. Significant decrease in latency was also found in the NTG control (*p* < 0.001) and 0.2 mg/kg THC (*p* < 0.05) groups compared to the TG control group in Block 4 Trial 5. (**D**) Significant decrease in the number of errors in 0.02 mg/kg THC (*p* < 0.01) and 0.2 mg/kg THC (*p* < 0.001) groups compared with the TG control group in Block 5 Trial 5. Significant decrease in the number of errors was also found in the NTG control (*p* < 0.001) and 0.2 mg/kg THC (*p* < 0.01) groups compared with the TG control group in Block 4 Trial 5. Data are expressed as mean ± SD. SD is denoted by the error bars. Comparison of mean latency and number of errors between female and male animals in individual study groups was made by multiple *t*-tests with correction for multiple comparisons using the Holm–Sidak method. Comparison of mean latency and number of errors among different study groups were made using one-way ANOVA with post hoc Bonferroni’s multiple comparisons test. * *p* < 0.05, ** *p* < 0.01 and *** *p* < 0.001 compared between the control NTG (*N* = 7), control TG (*N* = 6), 0.02 mg/kg THC (*N* = 8), and 0.2 mg/kg THC (*N* = 6) groups.

**Figure 5 ijms-23-02757-f005:**
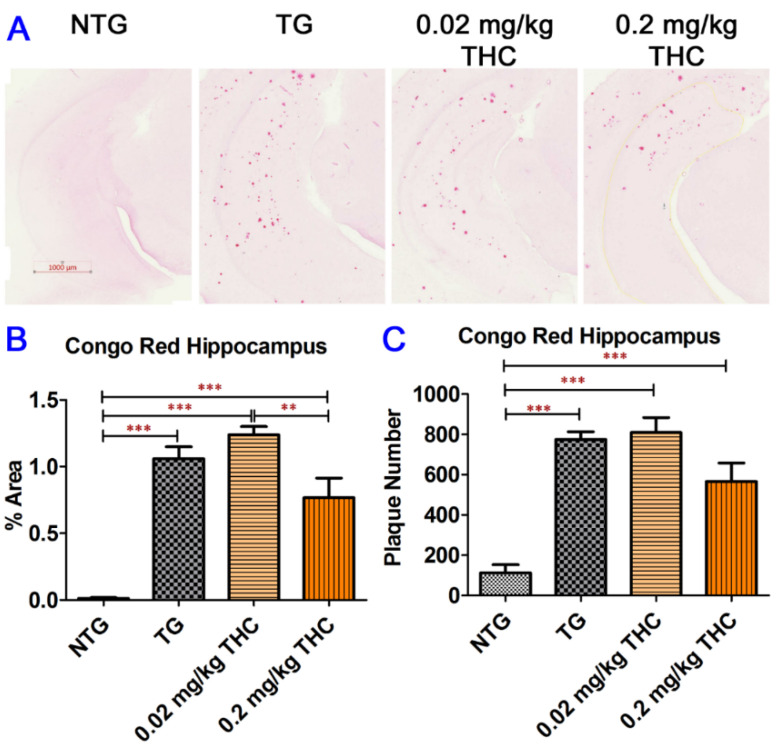
Congo Red staining of Aβ plaques in mouse hippocampi. (**A**) Representative Congo Red staining images acquired under light microscopy. (**B**) Quantification of Congo red staining shown as the percentage of Congo red-positive area compared to the hippocampus tissue area per field. (**C**) Quantification of Congo red staining shown as the number of Congo red stained plague in the hippocampus area. The non-transgenic (NTG) mice had significantly fewer Aβ plaques than all the APP/PS1 transgenic (TG) mice regardless of treatment (*p* < 0.001 for all). No significant differences in Aβ plaque area and number of Aβ plaques were found between the vehicle control and 0.02 or 0.2 mg/kg THC treated APP/PS1 mice. However, the Aβ plaque area in hippocampi sections of APP/PS1 mice treated with 0.2 mg/kg THC was significantly lower than those treated with 0.02 mg/kg THC (*p* < 0.01). Data are expressed as mean ± SD (*N* = 7 for the control NTG group, *N* = 6 for the control TG, and 0.2 mg/kg THC groups, and *N* = 8 for the 0.02 mg/kg THC group). Error bars denote the SD. ** *p* < 0.01 and *** *p* < 0.001 compared between the control NTG mice, control APP/PS1 mice, and APP/PS1 mice treated with 0.02 and 0.2 mg/kg THC using one-way ANOVA followed by the Tukey–Kramer post hoc multiple comparison test.

**Figure 6 ijms-23-02757-f006:**
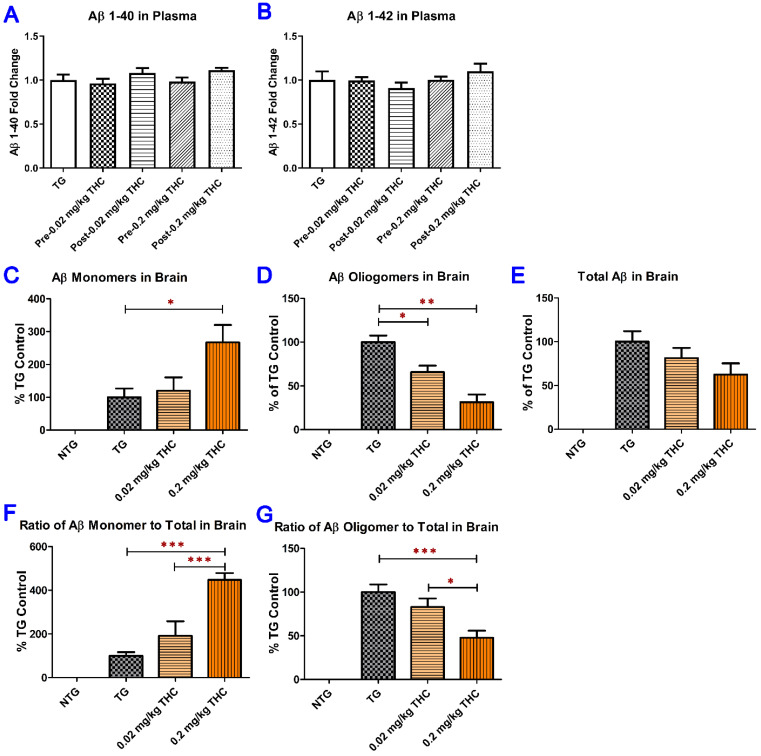
Determination of both soluble and insoluble Aβ1–40 (**A**) and Aβ1–42 (**B**) levels in the plasma using ELISA. Plasma samples were collected from the APP/PS1 mice before the start and after the 3-month THC treatment. No significant difference in soluble and insoluble Aβ1–40 and Aβ1–42 levels in plasma was found among all treatment groups and between the baseline and post-treatment levels (*N* = 7 for the control NTG group, *N* = 6 for the control TG and 0.2 mg/kg THC groups, and *N* = 8 for the 0.02 mg/kg THC group). Determination of Aβ monomers (**C**), Aβ oligomers (**D**), and total Aβ (**E**) in mouse brain tissue using the semi-quantitative western blotting and the total Aβ normalized Aβ monomer (**F**) and oligomer levels (**G**). The total Aβ normalized Aβ monomer (or oligomer) level was calculated as the ratio of Aβ monomer (or oligomer) level to total Aβ level. Treatment with 0.2 mg/kg THC significantly increased the Aβ monomer level and decreased Aβ oligomer level compared with the control treatment in TG mice. Data are expressed as mean ± SD (*N* = 6 for each study group). SD is denoted by the error bars. * *p* < 0.05, ** *p* < 0.01, and *** *p* < 0.001 compared between the control NTG mice, control APP/PS1 mice, and APP/PS1 mice treated with 0.02 and 0.2 mg/kg THC using one-way ANOVA followed by the Tukey–Kramer post hoc multiple comparison test.

**Figure 7 ijms-23-02757-f007:**
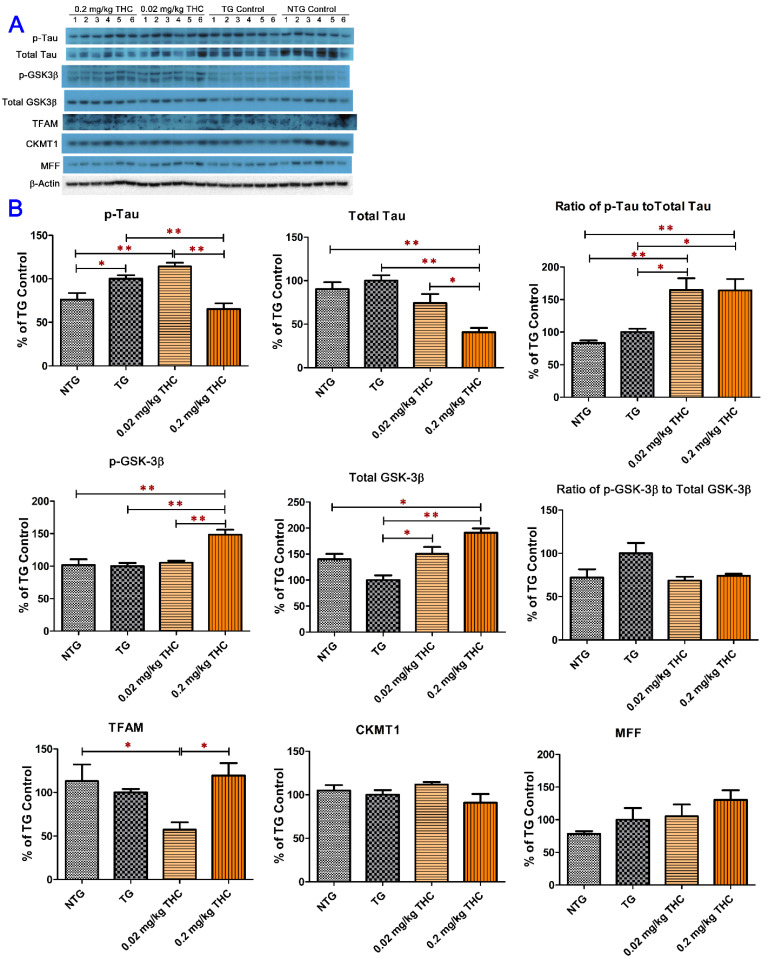
Evaluation of the effect of THC on the protein expression of phospho-tau, total tau, phospho-GSK-3β, total GSK-3β, TFAM, CKMT1, and MFF in brain homogenates using western blot analysis. (**A**) Western blot images of the expression of total and phosphorylated Tau and GSK-3β, TFAM, CKMT1, and MFF proteins in individual brain homogenate samples collected from the control NTG (*N* = 6), control TG (*N* = 6), 0.02 mg/kg (*N* = 6), and 0.2 mg/kg (*N* = 6) THC treatment groups. Detection of β-actin was used to ensure equal sample loading per lane. (**B**) Relative immunoreactive band intensities are expressed as percent change over the average signal value in the control TG mouse brain homogenates. THC treatment at 0.2 mg/kg significantly decreased the expression levels of phospho-tau and total Tau and increased the expression levels of phospho-GSK3β and total GSK3β compared with the vehicle treatment in APP/PS1 mice. THC treatment at either 0.02 mg/kg or 0.2 mg/kg had no significant effect on the protein levels of TFAM, CKMT1, and MFF in brain homogenates. Data are presented as mean ± SD (*N* = 6 for each study group). SD is denoted by the error bars. * *p* < 0.05 and ** *p* < 0.01 compared between the control NTG mice, control APP/PS1 mice, and APP/PS1 mice treatment with 0.02 and 0.2 mg/kg THC using one-way ANOVA followed by Tukey–Kramer post hoc multiple comparison test.

**Figure 8 ijms-23-02757-f008:**
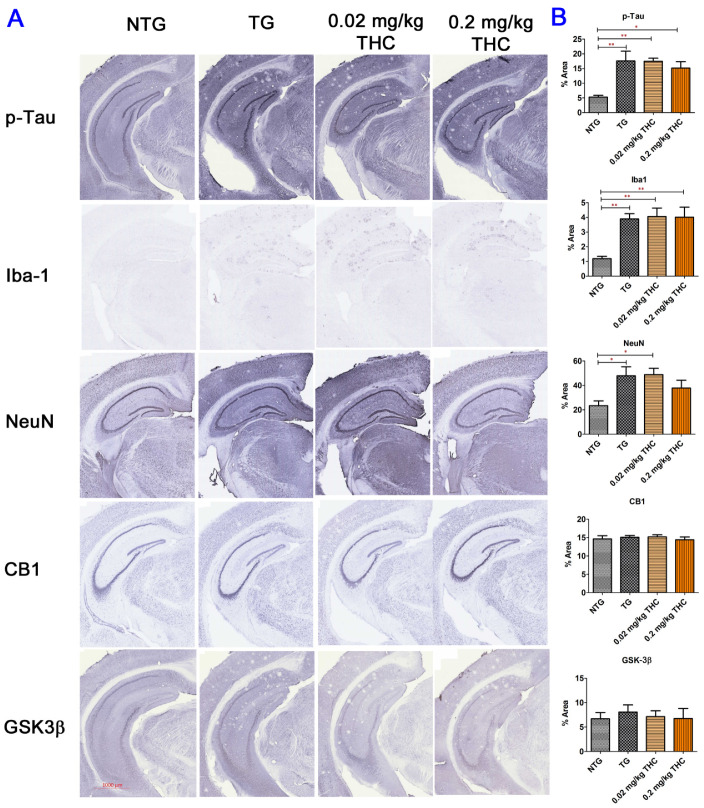
Immunohistochemical (IHC) analysis of molecular markers associated with the neuropathologic change in AD. (**A**) Representative IHC images for p-Tau, Iba-1, NeuN, CB1, and GSK3β in brain sections. (**B**) Quantification of IHC staining of for p-Tau, Iba-1, NeuN, CB1, and GSK3β in brain sections. No statistically significant difference in the expression of phospho-Tau, Iba1, CB1, and GSK-3β and the NeuN area was observed between the vehicle and THC treatment in APP/PS1 mice, suggesting the limited effect of THC on reversing the neuropathologic change in AD. Data are expressed as mean ± SD (*N* = 7 for the control NTG group, *N* = 6 for the control TG and 0.2 mg/kg THC groups, and *N* = 8 for the 0.02 mg/kg THC group). SD is denoted by the error bars. * *p* < 0.05 and ** *p* < 0.01 compared between the control NTG mice, control APP/PS1 mice, and APP/PS1 mice treatment with 0.02 and 0.2 mg/kg THC using one-way ANOVA followed by the Tukey–Kramer post hoc multiple comparison test.

**Figure 9 ijms-23-02757-f009:**
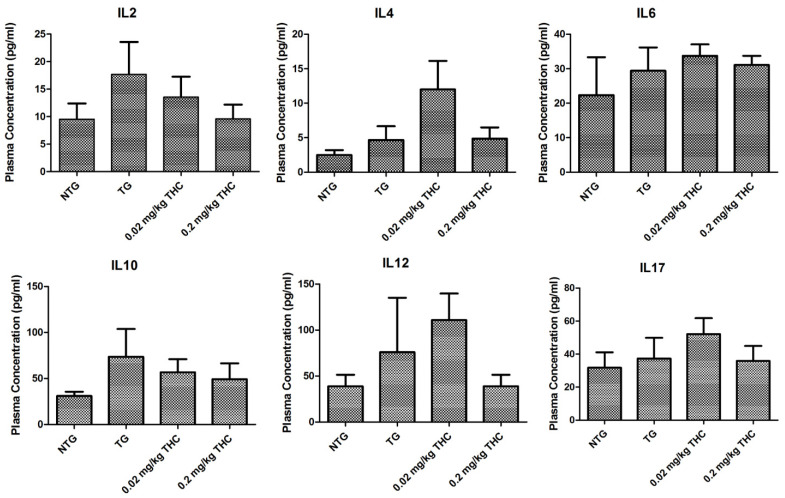
Evaluation of the effect of THC treatment on the plasma cytokine levels. Control NTG mice (*N* = 7) were given IP administration of PBS, while APP/PS1 mice were given IP administration of PBS (*N* = 6), 0.02 mg/kg THC (*N* = 8), or 0.2 mg/kg THC (*N* = 6) once every other day for three months. Plasma cytokine levels were determined using ELISA after the treatments were completed. No statistically significant difference in the plasma levels of any cytokine was observed between any study group. Data are presented as mean ± SD. Statistical analysis was conducted using one-way ANOVA followed by the Tukey–Kramer post hoc multiple comparison test.

**Figure 10 ijms-23-02757-f010:**
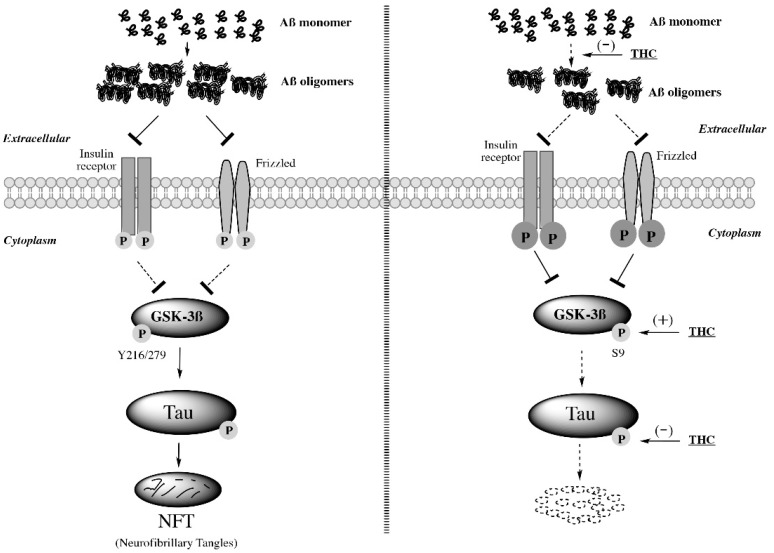
Inhibitory effect of low-dose THC treatment on Aβ aggregation, GSK-3β activity, and tau phosphorylation in the brain. Low-dose THC prevents Aβ monomers from forming Aβ oligomers and alleviates the antagonistic effect of Aβ oligomers on insulin receptors and Frizzled so that insulin receptors and Frizzled are able to downregulate GSK-3β activity. Moreover, low-dose THC may decrease GSK-3β activity directly by increasing the Ser9 phosphorylation of GSK-3β. Since the increased GSK-3β activity promotes tau phosphorylation, it is likely that the inhibitory effect of THC on tau phosphorylation is in part attributable to the THC-induced Ser9 phosphorylation of GSK-3β.

## Data Availability

The data presented in this study are available on request from the corresponding authors.

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
