# Peer review of "Low-Dose Delta-9-Tetrahydrocannabinol as Beneficial Treatment for Aged APP/PS1 Mice"

_ijms, 2022, doi:10.3390/ijms23052757_

Round 1
Reviewer 1 Report
I have to admit that I was impressed with the study. However, considering the conclusion, I would suggest to give an additional rationale for the used doses of THC as the "low" doses and the range of the used "low" doses. Likewise, considering ref 40 - I would suggest more detailed specification of the "high" doses.
Hope this would be no problem.
Author Response
Reviewer comments:
I have to admit that I was impressed with the study. However, considering the conclusion, I would suggest to give an additional rationale for the used doses of THC as the "low" doses and the range of the used "low" doses. Likewise, considering ref 40 - I would suggest more detailed specification of the "high" doses.
Hope this would be no problem.
Reply: We appreciate the reviewer's positive comments. As compared with the THC doses (1 and 3 mg/kg/d) used in the two recent studies by Zimmer's group, THC doses used in this study (0.02 and 0.2 mg/kg once every other day) were much lower. We have revised several sentences in the manuscript as follows to highlight the relatively low dose as compared with the documented studies and THC doses that could induce acute toxic responses.
The sentence in the third paragraph of the Introduction section (Line 96 - 98) has been changed to: "Two recent studies by Zimmer's group demonstrated that treatment with THC at doses of 1 and 3 mg/kg/d for 28 consecutive days significantly improved the spatial learning performance of 12 and 18 month-old C57BL6/J mice, whereas a 1:1 combination of THC and CBD (1 mg/kg/d for each) failed to achieve the same effect".
The sentences in the first paragraph of the Discussion section (Line 426-429) have been changed to: "... planning [40]. In mice, THC doses inducing acute intoxication that increased spontaneous anxiety-like and locomotor behavior were found to be 1 mg/kg for the intravenous administration and 10 mg/kg for the intraperitoneal administration. In this regard, the THC doses used in this study were much lower than the dose that could induce the anxiety-like behavior in mice. Contrary to the common belief, a handful of preclinical studies have revealed the beneficial effect of in vivo THC treatment at doses ranging of 0.002 and 3 mg/kg against the age-related decline in cognitive performance of aged mice. Unlike other documented in vivo studies, the present study was carried out in aged APP/PS1 mice, and the animals were treated with 0.02 and 0.2 mg/kg of THC once every other day, which were lower than the doses of 1 and 3 mg/kg/d reported by Zimmer's group. Result of our study ...".
Reviewer 2 Report
This manuscript presented significant results showing the safety and efficacy of THC in pre-clinical AD models. The authors have used various endpoints, e.g. behavior and pathological analyses, to support the potential efficacy of THC in ameliorating AD-related behavioral deficits, Aß- and Tau-related pathological changes.
My major concerns are:
- In Fig. 2, it is not clear why the authors did not study the combination of THC and CBD at 10nM. It should also be describe why THC and CBD combination need to be studies as all following experiments did not study their combination.
- In Fig. 5B, the authors wrote that "Although treatment with 0.02 mg/kg of THC appeared to decrease the Aβ plaque areas and number of Aβ plaques by 28% and 27%, respectively, as compared with the vehicle treatment, the difference was not statistically significant (Figure 5B and 5C)." This is consistent with the results shown in Fig. 5B. It looks like that 0.2 mg/kg dose group show the reduction of Aβ plaques.
- In Fig. 6C, it may be helpful to also show the levels of Aß monomer and oligomer normalized to total Aß levels.
- Fig. 7 and 8 are the most confusing results. Fig. 7A shows IHC staining results, however, Fig. 7B shows the quantification of western blotting analysis. Fig. 8 shows the quantification of IHC results. These two figures need to be revised, and the western blotting results should also been shown instead of being shown in the supplemental figure.
- In Fig. 9, the authors evaluated the effect of THC treatment on plasma cytokine levels. Their conclusion is that THC treatment at 0.02 and 0.2 mg/kg both have no effects on plasma levels of these cytokines. It is unclear why their levels in the brain tissues are not analyzed, especially IL2 and IL6 that are known to be associated with neuroinflammatory responses and AD pathogenesis.
Author Response
Reviewer 2 Comments:
This manuscript presented significant results showing the safety and efficacy of THC in pre-clinical AD models. The authors have used various endpoints, e.g. behavior and pathological analyses, to support the potential efficacy of THC in ameliorating AD-related behavioral deficits, Aß and Tau-related pathological changes.
My major concerns are:
- In Fig. 2, it is not clear why the authors did not study the combination of THC and CBD at 10nM. It should also be described why THC and CBD combination need to be studies as all following experiments did not study their combination.
Reply: In the pilot study, we tested THC and CBD alone at concentrations of 10 nM and lower and THC + CBD at 10 nM but did not observe any statistical significance among vehicle control and treatment groups. Therefore, in the subsequent study, the doses for THC and CBD alone were set at 10 and 100 nM and that for THC + CBD was set at 100 nM for each. We have added the following sentences to Section 2.2 (Line 125) address the concern: "Early studies have demonstrated that CBD treatment significantly reduced Aß1-40 production in vitro [14] while CBD-THC combination decreased soluble Aß1-42 levels in vivo [21]. In this regard, treatment with CBD alone and in combination with THC were included as positive controls in the in vitro study to examine if the inhibitory effect of THC treatment alone on Aß production in N2a/APPswe cells was comparable to that of CBD alone and CBD + THC".
- In Fig. 5B, the authors wrote that "Although treatment with 0.02 mg/kg of THC appeared to decrease the Aβ plaque areas and number of Aß plaques by 28% and 27%, respectively, as compared with the vehicle treatment, the difference was not statistically significant (Figure 5B and 5C)." This is consistent with the results shown in Fig. 5B. It looks like that 0.2 mg/kg dose group show the reduction of Aß plaques.
Reply: We appreciate the reviewer for spotting the typo. It should be 0.2 mg/kg instead of 0.02 mg/kg. We have corrected the typo in Section 2.4 (Line 267).
- In Fig. 6C, it may be helpful to also show the levels of Aß monomer and oligomer normalized to total Aß levels.
Reply: We have added two more plots to Fig. 6 to show the total Aß normalized Aß monomer (Fig. 6F) and oligomer (Fig. 6G) levels and included the following description in Section 2.5 (Line 305): "The significant effect of THC treatment at 0.2 mg/kg on Aß levels in the brain was also demonstrated by the fact that the total Aß normalized Aß monomer level in the 0.2 mg/kg THC group was significantly lower than those in the control and 0.02 mg/kg THC groups (P < 0.001 for both. Figure 6F), whereas the total Aß normalized Aß oligomer level in the 0.2 mg/kg THC group was significantly higher than those in the control (P < 0.001) and 0.02 mg/kg THC groups (P < 0.05. Figure 6G)".
- Fig. 7 and 8 are the most confusing results. Fig. 7A shows IHC staining results, however, Fig. 7B shows the quantification of western blotting analysis. Fig. 8 shows the quantification of IHC results. These two figures need to be revised, and the western blotting results should also been shown instead of being shown in the supplemental figure.
Reply: We sincerely apologize for our oversight of the embarrassing mix-up of Fig.7 and Fig.8. We have swapped Fig. 7 and Fig. 8 and revised the figure legends to match respective sections in the manuscript.
- In Fig. 9, the authors evaluated the effect of THC treatment on plasma cytokine levels. Their conclusion is that THC treatment at 0.02 and 0.2 mg/kg both have no effects on plasma levels of these cytokines. It is unclear why their levels in the brain tissues are not analyzed, especially IL2 and IL6 that are known to be associated with neuroinflammatory responses and AD pathogenesis.
Reply: We agree with the reviewer that neuroinflammation is a prominent factor contributing to the development of AD. Although we did not directly quantify the cytokine and chemokine levels in mouse brain tissues due to the limited brain samples collected from the in vivo study, we looked into the immunohistochemical evidence of microglial activation in plaques associated with fibrillar Aß deposits (Congo red staining), Tau hyperphosphorylation and GSK3β activation in the absence and presence of THC treatment, as well as the cytokine levels in plasma. The results were promising but not definitive. We realized that the transgenic APP/PS1 mouse AD model used in this study might not be optimal to evaluate the possible effect of THC on neuroinflammation in that neuroinflammation often occurs as a secondary response to sustain Aß overproduction and deposition in traditional animal AD models, and thus is incomplete and less severe as compared with the neuroinflammation in patients with AD (Krstic D and Knuesel I, Nat Rev Neurol. 2013;9(1):25-34). In our future study, we will use a different transgenic mouse model of AD along with certain immune challenge-base animal models to examine the anti-inflammatory effect of THC in AD and associated mechanisms."
Reviewer 3 Report
The authors study the effect of the low-dose delta-9-Tetrahydrocannabinol Beneficial on Aged APP/PS1 Mice. I find this work very interesting ad I recommend the publication of the manuscript after minor revision.
Lines 37-39: Although THC has exactly the same chemical formula as CBD, i.e., C21H30O2, there is a slight difference in their atomic arrangement in that THC contains a cyclic ring, whereas CBD contains a hydroxyl group [1,2]. Please correct this sentence the chemical formula of CBD C21H30O2 is different from that of THC C21H26O2. Moreover, the phrase will be clearer if you insert a figure with the chemical structure of both compounds.
Lines 128-131: As shown in 128 Figure 2A, treatment with 10 nM THC, 100 nM THC, 100 nM CBD and 100 nM of THC 129 and CBD in combination for 24 hours significantly decreased the Aβ1-40 production in 130 N2a/APPswe cells by 23% (P < 0.05), 27% (P < 0.01), 21% (P < 0.05) and 32% (P < 0.001), 131 respectively. In this sentence you do not comment the effect of 10 nM of CBD. Please add few words.
The significativity of the test in Fig. 2 (C and D) has not been provided, even if in the text it has been stated that "treatments with 10 and 100 nM of THC for 42 hours were able to de-139 crease the Aβ1-42 production in N2a/APPswe cells by 16% and 24%, respectively". Is this decrease statistically significant?
The authors should explain why the Aß 1-40 levels at 42 hours are in some cases higher than the control. Does this mean that at this concentration it is possible to observe an increase of Aß 1-40 production in response of the early decrease at 24 hours? The combination of CBD and THC doesn't seem to do the same.
The 0.2 mg/kg of THC seems to decrease significantly the amyloid plaques area compared to 0.02 mg/kg. The assessment of Aß levels in plasma showed that THC seems to increase the monomer plasmatic concentration. A question arises if this increase is due to inhibition of aggregation or amyloid disaggregation. The authors should assess the possible mechanism by performing some biophysical assays such as fluorescence spectroscopy.
The hypothesis that the THC is able to reduce the oligomeric amount seems to be a fast conclusion. The fact that the amount of monomer increases and the oligomers decrease is not a clear correlation of oligomer inhibition. It could happen that the aggregation process is inhibited by THC and reduce the amount of oligomeric toxic species (or different oligomers are there but not detectable by ELISA test) or it might be due to disaggregation process which brings to monomers bypassing the oligomeric species. This point should be point out by other biophysical assays.
Paragraph 2.6 and Fig 7 should be checked. The authors sometimes write "significantly relevant" but this is not clear on Fig. 7 (especially for phospho-GSK). Please check that there is a clear correspondence between the Fig. 7 and paragraph 2.6.
Paragraph 2.8, lines 406-407: the sentence is not complete.
Lines 439-440: the sentence should be revised. This mechanism has not be totally explored.
Lines 448-449: this is just an hypothesis of the authors. The sentence should be written as "it might be possible that..."
Lines 451-453: the reduce of amyloid deposit is not significantly relevant compared to the control. An increase level of monomers is not representative of a stabilization of monomer.
Lines 460-462: It's clear for oligomers but not really from Fig. 7 for p-tau.
Lines 469-472: is it really significant? From Fig. 7 it doesn't seem, especially for GSK-3B.
Author Response
Reviewer 3 Comments:
The authors study the effect of the low-dose delta-9-Tetrahydrocannabinol Beneficial on Aged APP/PS1 Mice. I find this work very interesting ad I recommend the publication of the manuscript after minor revision.
Reply: We appreciate the reviewer's positive comments.
Lines 37-39: Although THC has exactly the same chemical formula as CBD, i.e., C21H30O2, there is a slight difference in their atomic arrangement in that THC contains a cyclic ring, whereas CBD contains a hydroxyl group [1,2]. Please correct this sentence the chemical formula of CBD C21H30O2 is different from that of THC C21H26O2. Moreover, the phrase will be clearer if you insert a figure with the chemical structure of both compounds.
Reply: We double-checked the molecular formulas of CBD and THC and confirmed that both compounds have the same formula of C21H30O2. We have included the structures of THC and CBD as Figure S1 in Supplemental Materials.
Lines 128-131: As shown in 128 Figure 2A, treatment with 10 nM THC, 100 nM THC, 100 nM CBD and 100 nM of THC 129 and CBD in combination for 24 hours significantly decreased the Aß1-40 production in 130 N2a/APPswe cells by 23% (P < 0.05), 27% (P < 0.01), 21% (P < 0.05) and 32% (P < 0.001), 131 respectively. In this sentence you do not comment the effect of 10 nM of CBD. Please add few words.
Reply: The following sentence has been added to Section 2.2 (Line 131): "The mean Aß1-40 level in N2a/APPswe cells treated with 10 nM CBD was not significantly different from that of the vehicle control (Figure 2A.)."
The significativity of the test in Fig. 2 (C and D) has not been provided, even if in the text it has been stated that "treatments with 10 and 100 nM of THC for 42 hours were able to de-139 crease the Aß1-42 production in N2a/APPswe cells by 16% and 24%, respectively". Is this decrease statistically significant?
Reply: We have added "P > 0.05 for all" in Section 2.2 (Line 138) to indicate the difference observed in the cellular levels of Aß1-40 and Aß1-42 was not statistically significant among all treatment groups.
The authors should explain why the Aß 1-40 levels at 42 hours are in some cases higher than the control. Does this mean that at this concentration it is possible to observe an increase of Aß 1-40 production in response of the early decrease at 24 hours? The combination of CBD and THC doesn't seem to do the same.
Reply: Since the results of statistical analysis indicated that the difference was not statically significant, we cannot speculate about why this effect might have occurred.
The 0.2 mg/kg of THC seems to decrease significantly the amyloid plaques area compared to 0.02 mg/kg. The assessment of Aß levels in plasma showed that THC seems to increase the monomer plasmatic concentration. A question arises if this increase is due to inhibition of aggregation or amyloid disaggregation. The authors should assess the possible mechanism by performing some biophysical assays such as fluorescence spectroscopy.
Reply: We appreciate the reviewer's insightful comments. Our speculation was that it could be due to both. In our previous study (Cao et al JAD 2014, reference #24), we demonstrated that THC inhibited Aß1-40 aggregation in vitro using the thioflavin T fluorescence assay. Â We will look into the possibility of using a fluorescence correlation spectroscopy to monitor the THC effect on Aß1-40 aggregation in vivo.
The hypothesis that the THC is able to reduce the oligomeric amount seems to be a fast conclusion. The fact that the amount of monomer increases and the oligomers decrease is not a clear correlation of oligomer inhibition. It could happen that the aggregation process is inhibited by THC and reduce the amount of oligomeric toxic species (or different oligomers are there but not detectable by ELISA test) or it might be due to disaggregation process which brings to monomers bypassing the oligomeric species. This point should be point out by other biophysical assays.
Reply: We agree with the reviewer totally. We have revised the last sentence of the second paragraph in the Discussion section (Line 455-453) as follows: "Given the overall observations in this study as well as the results of our previous thioflavin T fluorescence assay showing the inhibitory effect of THC on Aß1-40 aggregation in vitro [24], it is speculated that low-dose THC treatment has the potential to reduce amyloid deposits in the brain by preventing the aggregation of Aß monomers and the formation of Aβ oligomers. Further studies are warranted to investigate the effect of THC on the polymerization process of Aß through direct monitoring of the Aß monomer to oligomer transition in vivo using fiber-based fluorescence correlation stereoscopy [44]".
Paragraph 2.6 and Fig 7 should be checked. The authors sometimes write "significantly relevant" but this is not clear on Fig. 7 (especially for phospho-GSK). Please check that there is a clear correspondence between the Fig. 7 and paragraph 2.6.
Reply: We sincerely apologize for our oversight of the embarrassing mix-up of Fig.7 and Fig.8. We have swapped Fig. 7 and Fig. 8 and revised the figure legends to match respective sections in the manuscript.
Paragraph 2.8, lines 406-407: the sentence is not complete.
Reply: The sentence in Section 2.8 (Line 406) has been changed to "However, the difference between APP/PS1 control group and either of the THC treatment groups was not statistically significant (P > 0.05 for all. Figure 9)".
Lines 439-440: the sentence should be revised. This mechanism has not be totally explored.
Reply: Although accumulating data from our previous in vitro study (Cao et al JAD 2014) and the current in vivo study have all pointed to the ability of THC to inhibit Aß production and aggregation as a potential mechanism underlying its beneficial effect on AD, we understand that more work needs to be done to delineate the mechanism moving forward. As per the reviewer's suggestion, the sentence in the second paragraph of the Discussion section (Line 439-440) has been changed to "The observed ability of low-dose THC treatment to improve the spatial memory performance of aged APP/PS1 mice coincided with its inhibitory effect on Aß production and aggregation, implicating that THC might reduce Aß toxicity by targeting its polymerization in the brain."
Lines 448-449: this is just a hypothesis of the authors. The sentence should be written as "it might be possible that..."
Reply: The sentence in the second paragraph of the Discussion section (Line 448-449) has been changed to "Based on the finding, it might be possible that an extended THC treatment period would result in a significant reduction of the amyloid deposits in the brain" .
Lines 451-453: the reduce of amyloid deposit is not significantly relevant compared to the control. An increase level of monomers is not representative of a stabilization of monomer.
Reply: The last sentence of the second paragraph in the Discussion section has been revised to "Given the overall observations in this study as well as the results of our previous thioflavin T fluorescence assay showing the inhibitory effect of THC on Aβ1-40 aggregation in vitro [24], it is speculated that low-dose THC treatment has the potential to reduce amyloid deposits in the brain by preventing the aggregation of Aß monomers and the formation of Aß oligomers. Further studies are warranted to investigate the effect of THC on the polymerization process of Aß through direct monitoring of the Aß monomer to oligomer transition in vivo using fiber-based fluorescence correlation stereoscopy [44]".
Lines 460-462: It's clear for oligomers but not really from Fig. 7 for p-tau.
Reply: As we mentioned previously, there was a mix-up of Fig.7 and Fig.8. We have swapped Fig. 7 and Fig. 8 and revised the figure legends to match the respective sections in the manuscript. The significant changes were meant for the Western blot results.
Lines 469-472: is it really significant? From Fig. 7 it doesn't seem, especially for GSK-3B.
Reply: Same as above. The significant changes were meant for the Western blot results.